

# Effects of knee sleeves on coordination of lower-limb segments in healthy adults during level walking and one-leg hopping

Chang-Yong Ko, Yunhee Chang, Bora Jeong, Sungjae Kang, Jeicheong Ryu and Gyoosuk Kim

Research Team, Rehabilitation Engineering Research Institute, Incheon, Republic of Korea

## ABSTRACT

The evaluation of multisegment coordination is important in gaining a better understanding of the gait and physical activities in humans. Therefore, this study aims to verify whether the use of knee sleeves affects the coordination of lower-limb segments during level walking and one-leg hopping. Eleven healthy male adults participated in this study. They were asked to walk 10 m on a level ground and perform one-leg hops with and without a knee sleeve. The segment angles and the response velocities of the thigh, shank, and foot were measured and calculated by using a motion analysis system. The phases between the segment angle and the velocity were then calculated. Moreover, the continuous relative phase (CRP) was calculated as the phase of the distal segment subtracted from the phase of the proximal segment and denoted as CRPTS (thigh–shank), CRPSF (shank–foot), and CRPTF (thigh–foot). The root mean square (RMS) values were used to evaluate the in-phase or out-of-phase states, while the standard deviation (SD) values were utilized to evaluate the variability in the stance and swing phases during level walking and in the preflight, flight, and landing phases during one-leg hopping. The walking velocity and the flight time improved when the knee sleeve was worn ($p < 0.05$). The segment angles of the thigh and shank also changed when the knee sleeve was worn during level walking and one-leg hopping. The RMS values of CRPTS and CRPSF in the stance phase and the RMS values of CRPSF in the preflight and landing phases changed ($p < 0.05$ in all cases). Moreover, the SD values of CRPTS in the landing phase and the SD values of CRPSF in the preflight and landing phases increased ($p < 0.05$ in all cases). These results indicated that wearing a knee sleeve caused changes in segment kinematics and coordination.

Corresponding author
Chang-Yong Ko,
monamicyko@gmail.com,
cyko@kcomwel.or.kr

# INTRODUCTION

Knee sleeves are widely used to treat and prevent knee problems, such as knee osteoarthritis and pain, in both occupational and athletic settings. *Beaudreuil et al. (2009)* stated the clinical effectiveness of knee sleeve for knee osteoarthritis. *Hrnack & Barber (2014)* suggested that knee brace was one of the effective management methods of the pain of knee osteoarthritis. Some studies investigated the alterations in proprioception in humans

wearing knee sleeves. *Van Tiggelen, Coorevits & Witvrouw (2008)* showed the positive effects of neoprene knee sleeves in overcoming the deficit in knee proprioception caused by muscle fatigue. Meanwhile, *Herrington, Simmonds & Hatcher (2005)* and *Barrett (2003)* showed that the sense of the knee joint position improved in healthy subjects and adolescent female athletes wearing neoprene knee sleeves.

Proprioception can affect the functional capacity and performance associated with motor control (*Hettich et al., 2014*; *Riemann & Lephart, 2002*; *Serrien et al., 2001*; *Tunik et al., 2003*). Therefore, some studies evaluated the effects of using a knee sleeve on the functional capacity and performance. *Mortaza et al. (2012)* showed that using a knee sleeve did not have any negative effects on the performance of a single-leg vertical jump and a crossover hop in healthy adults. *Bryk et al. (2011)* found that immediate positive effects were seen in the functional capacity of osteoarthritis patients wearing knee sleeves. Several other researchers also evaluated the effects of using a knee sleeve on the kinematics and kinetics of walking or other physical activities. *Collins et al. (2014)* showed that during walking, the sagittal-plane knee kinematics and kinetics improved in patients with osteoarthritis when they wore knee sleeves. In addition, *Schween, Gehring & Gollhofer (2015)* showed that the use of knee sleeves had a positive effect on the frontal-plane knee kinematics and kinetics in osteoarthritis patients during walking.

However, the abovementioned studies evaluated the kinematics and kinetics of a single joint/segment. The coordination of multiple segments of the lower extremity is required to accomplish a complex task, such as human walking, or any other physical activity with precise endpoint control. Therefore, evaluating the biomechanical features of a single joint/segment using traditional kinematic analyses may be insufficient in understanding the gait and physical activities in humans, and evaluating the multi-segment coordination is required. The continuous relative phase (CRP) is one of the most common parameters used to evaluate multi-joint/segment coordination. Many studies assessed the CRP to evaluate pathological and/or abnormal gait (*Barela et al., 2000*; *Chiu et al., 2015*; *Chiu, Lu & Chou, 2010*; *Hamill, Palmer & Emmerik, 2012*; *Yi et al., 2016*). The CRP is calculated by monitoring the segment movement and its response velocity and thus it considers temporal and spatial patterns, which are important elements of motor coordination.

We hypothesized that a knee sleeve might alter joint or segment coordination based on the influences of a knee sleeve on the functional capacity and performance associated with the joint or segment kinematics (*Bryk et al., 2011*; *Collins et al., 2014*; *Mortaza et al., 2012*; *Schween, Gehring & Gollhofer, 2015*). However, to the best of our knowledge, no study has focused on the effects of using knee sleeves on joint or segment coordination. Therefore, we aim to verify whether the use of knee sleeves alters the coordination of the lower-limb segments during level walking and one-leg hopping.

## METHODS

### Subjects

Eleven male subjects with asymptomatic back, hip, knee, and ankle functions participated in this study. Their mean (±SD) age, height, and weight were 24.8 ± 2.8 years, 179.8 ± 8 cm, and 80.6 ± 15.5 kg, respectively. The study was approved by the Human Ethics Committee

of the Rehabilitation and Engineering Research Institute, Korea (RERE-IRB-20160721). Informed consent was obtained from all the subjects before the experiments.

## Experimental procedures

The experiments were monitored using a three-dimensional (3D) motion analysis system (Motion Analysis, Santa Rosa, CA, USA), which consisted of 12 infrared cameras (Raptor-4S; Motion Analysis, Santa Rosa, CA, USA), four force plates (600 × 900 mm, AMTI, USA), passive reflective markers, a data acquisition system, and a software package (Cortex Ver. 6.3).

We used Helen Hayes marker sets to place 19 12.5 mm reflective markers on the anatomical landmarks of the lower limb. The markers were attached on the sacrum, anterior superior iliac spine (ASIS, bilaterally), lateral femoral epicondyle (bilaterally), calcaneus and malleolus (bilaterally), 2nd metatarsal head (bilaterally), and lower lateral 1/3 surface of both shanks and thighs (bilaterally). The kinematic data of all the markers and analog signals of the force plates were sampled at 120 Hz using real-time software (Cortex Ver. 6.3; Motion Analysis, Santa Rosa, CA, USA). The anthropometric data of each subject, including height and weight, were measured prior to the experiments. The subjects were later instructed to repeatedly walk on a 10 m walkway for 10 min to induce a natural gait pattern and to provide an adaptation period with the knee sleeve. The subjects walked at their self-selected walking speeds (SSWS) along the gaitway. The marker 3D position data, the ground reaction forces and the real-time image data were identified and measured by real-time software during the gait analysis. The data on each marker were smoothed by Butterworth filters at 6 Hz (*Kim et al., 2017*).

We measured two activities in the experiments: walking and one-leg hop test. We chose the one-leg hop test because this task required a large amount of lower extremity coordination and has also been highly used to assess the lower extremity function (*Grindem et al., 2011*; *Van Uden et al., 2003*). The knee brace used in our study was a sleeve with a silicone patella pad (MEDI, Bad Homburg, Germany). The subjects wore the sleeve only on their knee of their preferred legs, all right knee in this study. The knee sleeve for each subject was determined according to the sizing chart of the sleeve and fitted according to the manufacturer's guideline. We analyzed the change in the spatiotemporal and CRP parameters during walking and one-leg hopping before and after the knee brace was worn. The initial contact and toe off for the gait and hopping test were detected by using zero ground reaction force and visual inspection (*Zeni, Richards & Higginson, 2008*).

All the subjects practiced walking before the actual gait analysis to induce a natural gait. They were then asked to walk at self-selected walking speed (*Bohannon, 1997*). We obtained five successful gait trials, where a clean foot strike on the force plate and a maintained self-selected walking speed (±5%) within was defined as a successful trial. We observed the force exerted on the force plate through the foot contact period to verify the successful gait trials. For the one-leg hop test, the subjects practiced hopping once or twice, such that they could achieve the maximum jumping distance using one leg. We measured the values from three trials, then obtained their mean value. The one-leg hop activity was divided into the preflight, flight, and landing phases (Fig. 1A). The flight phase represents

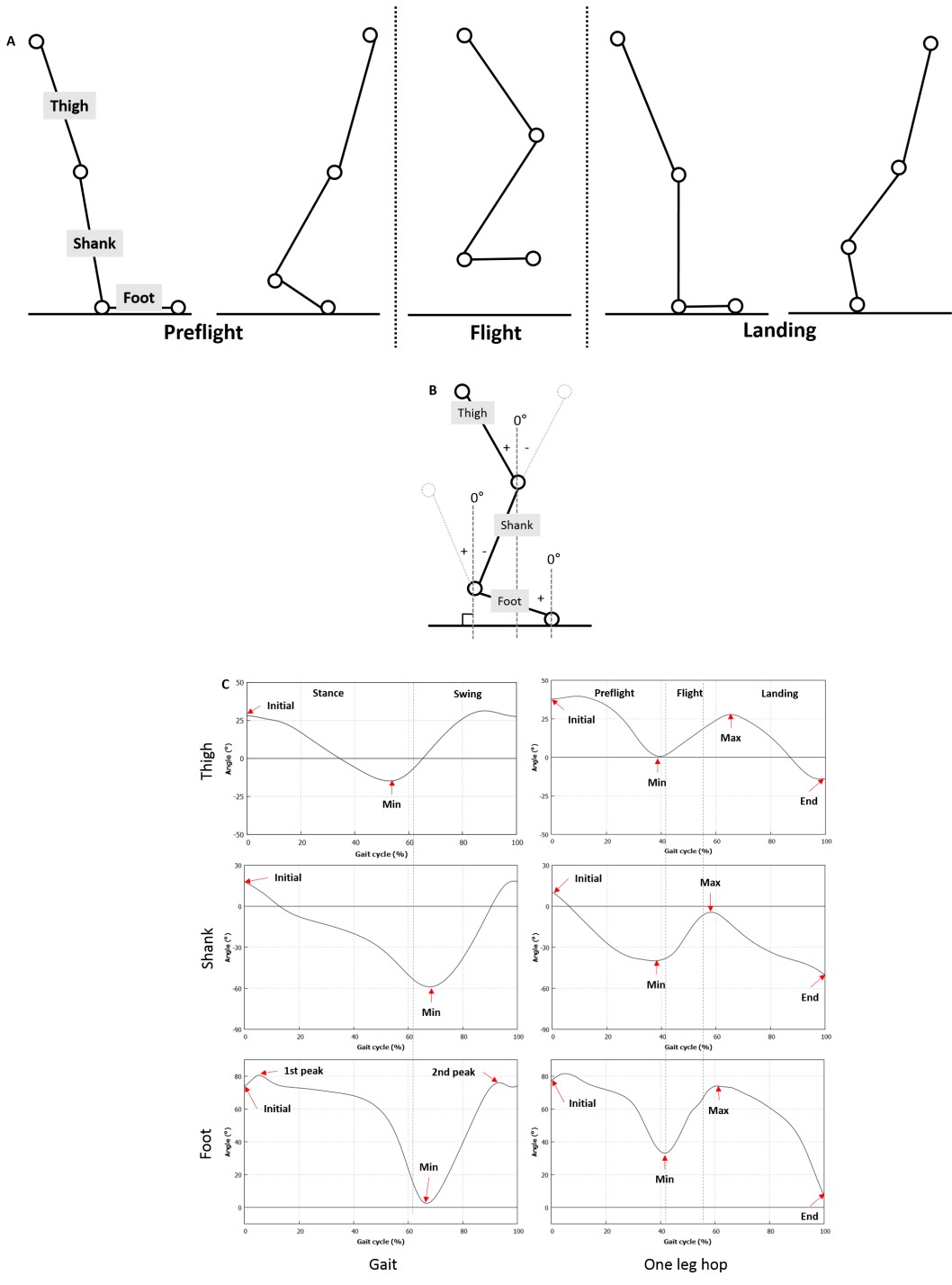

**Figure 1** **Definition of the segment angle and the phase and specific values of the segment angle.** (A) The one-leg hop activity is divided into the preflight, flight, and landing phases. (B) The sagittal angles of the thigh, shank, and foot with the vertical axis to the ground are calculated by using the marker data obtained from the motion analysis system. (C) Specific sections of the segment angle curves are defined.

the period, in which the maximum jumping distance was achieved using one leg, and the landing phase represented the period after the subject lands on the floor using the same leg. All the test procedures were randomly conducted by drawing lots.

## Measurement parameters

### Spatiotemporal parameters

A Cortex Ver. 6.3 was used to estimate the spatiotemporal parameters. The spatiotemporal parameters included walking velocity, cadence, step width, stride length, step length, and support time. These parameters were analyzed using the marker information, which contained data from a 3D space.

### Segment angle

A Cortex Ver. 6.3 was used to estimate the joint kinematics. The sagittal angles of the thigh, shank, and foot with the vertical axis to the ground were calculated by using the marker data obtained from the motion analysis system (Fig. 1B). The angle was 0° when perpendicular to the ground. The negative values of the angles of each segment indicated a clockwise direction to the reference axis, whereas the positive values implied a counter-clockwise direction. Figure 1C shows specific sections of the segment angle curves. In particular, the initial or minimum angle indicated angle at initial contact or push off, respectively.

### CRPs

The segment velocity corresponding to the segment angle was calculated and interpolated from 0% to 100%, including the stance and swing phases during a gait cycle and the preflight, flight, and landing phases during a one-leg hop. The angles and velocities were normalized between $-1$ and 1. The phase portraits of the normalized velocity ($y$-axis, normV) against the normalized angle ($x$-axis, norm$\theta$) during a gait or a one-leg hop test were then generated. The calculated phase was sometimes outside the range of negative 180° to positive 180°, resulting in the increase in discontinuity of the phase portraits. Unwrapping was performed by a multiple of 360° to maintain the continuity of the phase portraits. The CRP angle between the normalized velocity and the normalized angle was then calculated at each cycle as follows: $\text{CRP} = \tan^{-1} \frac{\text{normV}}{\text{norm}\theta}$. Subsequently, the average of the root mean square (RMS) and the standard deviation (SD) values were calculated over each phase: the stance and swing phases during a gait cycle and the preflight, flight, and landing phases during a one-leg hop. All the procedures were performed by using R (*R Core Team, 2017*), and signal (*Signal Developers, 2013*) and xlsx packages (*Dragulescu, 2014*).

## Statistical analysis

A paired $t$-test or a Wilcoxon $t$-test for non-uniformally distributed data was performed using SPSS v20 (IBM, USA) to verify the differences in the spatiotemporal parameters, segment angles, and RMS and SD values of CRPs between the cases with and without the knee sleeve. A Shapiro–Wilk test was also performed to evaluate the normality of the values. A $p$-value of less than 0.05 was considered significant.

**Table 1  Mean ± S.D. values of the spatiotemporal parameters.** Significant differences on the spatiotemporal parameters of the subjects with knee sleeve and those without are found during gait and one-leg hop.

|  | Parameter | No sleeve | Sleeve | p-value |
|---|---|---|---|---|
| Gait | Step width (cm) | 12.1 ± 2.3 | 12.9 ± 2.7 | 0.178 |
|  | Velocity (cm/s) | 136.0 ± 6.9 | 140.1 ± 7.4 | 0.078 |
|  | Stride length (cm) | 147.2 ± 10.2 | 149.8 ± 10.1 | 0.120 |
|  | Cadence | 110.9 ± 7.0 | 112.3 ± 4.9 | 0.104 |
|  | Stance (%) | 62.6 ± 1.7 | 62.3 ± 1.4 | 0.122 |
|  | Swing (%) | 37.4 ± 1.7 | 37.7 ± 1.4 | 0.122 |
| One-leg hop | Flight time (s) | 0.2 ± 0.0 | 0.3 ± 0.0 | 0.159 |
|  | Distance (cm) | 119.6 ± 8.9 | 117.9 ± 7.0 | 0.151 |

**Table 2  Segment angles.** The initial and minimum angles of the thigh and the shank during gait for the subjects with the sleeve are different from those without. The minimum angles of the thigh and the shank during the one-leg hop for the subjects with the sleeve are also different from those without.

|  |  | Gait |  |  | One-leg hop |  |  |
|---|---|---|---|---|---|---|---|
|  |  | No sleeve | With sleeve | p-value | No sleeve | With sleeve | p-value |
| Thigh | Initial | 28.2 ± 2.7 | 26.6 ± 3.0 | 0.003 | 37.8 ± 4.9 | 37.5 ± 6.3 | 0.389 |
|  | Min | −14.9 ± 2.4 | −17.6 ± 2.8 | 0.001 | −5.3 ± 4.0 | −8.8 ± 3.8 | 0.020 |
|  | Max | N/A | N/A |  | 33.2 ± 4.5 | 33.9 ± 4.4 | 0.311 |
| Shank | Initial | 17.8 ± 2.9 | 20.3 ± 2.7 | 0.001 | 10.3 ± 4.6 | 11.5 ± 8.8 | 0.269 |
|  | Min | −59.2 ± 2.6 | −56.4 ± 2.02 | <0.001 | −43.1 ± 4.4 | −37.4 ± 3.8 | <0.001 |
|  | Max | N/A | N/A | N/A | 5.0 ± 4.6 | 8.2 ± 4.0 | 0.009 |
| Foot | Initial | 73.9 ± 4.3 | 73.3 ± 4.4 | 0.139 | 77.3 ± 5.8 | 76.1 ± 6.4 | 0.227 |
|  | 1st peak | 81.7 ± 4.6 | 80.8 ± 4.3 | 0.113 | N/A | N/A | N/A |
|  | Min | 1.3 ± 5.7 | 1.3 ± 5.2 | 0.493 | 19.5 ± 6.0 | 18.5 ± 5.4 | 0.161 |
|  | 2nd peak | 78.3 ± 6.4 | 78.6 ± 6.2 | 0.292 | N/A | N/A | N/A |
|  | Max | N/A | N/A | N/A | 80.2 ± 8.3 | 79.0 ± 7.9 | 0.151 |

## RESULTS

Table 1 lists the spatiotemporal parameters. No differences were found in any of the spatiotemporal parameters ($p > 0.05$). Meanwhile, significant increases were observed in the flight time and the jump distance when the sleeve was worn ($p > 0.05$ in all cases) during the one-leg hop.

Table 2 and Fig. 2 show the angles of each segment. The initial and minimum angles of the thigh during level walking were lower for the subjects wore the sleeve ($p = 0.003$ and $p = 0.001$, respectively). The initial and minimum angles of the shank were higher for the the subjects wore the sleeve ($p = 0.001$ and $p < 0.001$, respectively). No significant differences were found in the other segment angles ($p > 0.05$ in all cases).

The minimum angles of the thigh for the subjects with the sleeve during the one-leg hop were lower than those without ($p = 0.020$). Meanwhile, the minimum angles of the shank for the subjects with the sleeve were higher than those without ($p < 0.001$). No significant differences were found in the other segment angles ($p > 0.05$ in all cases).
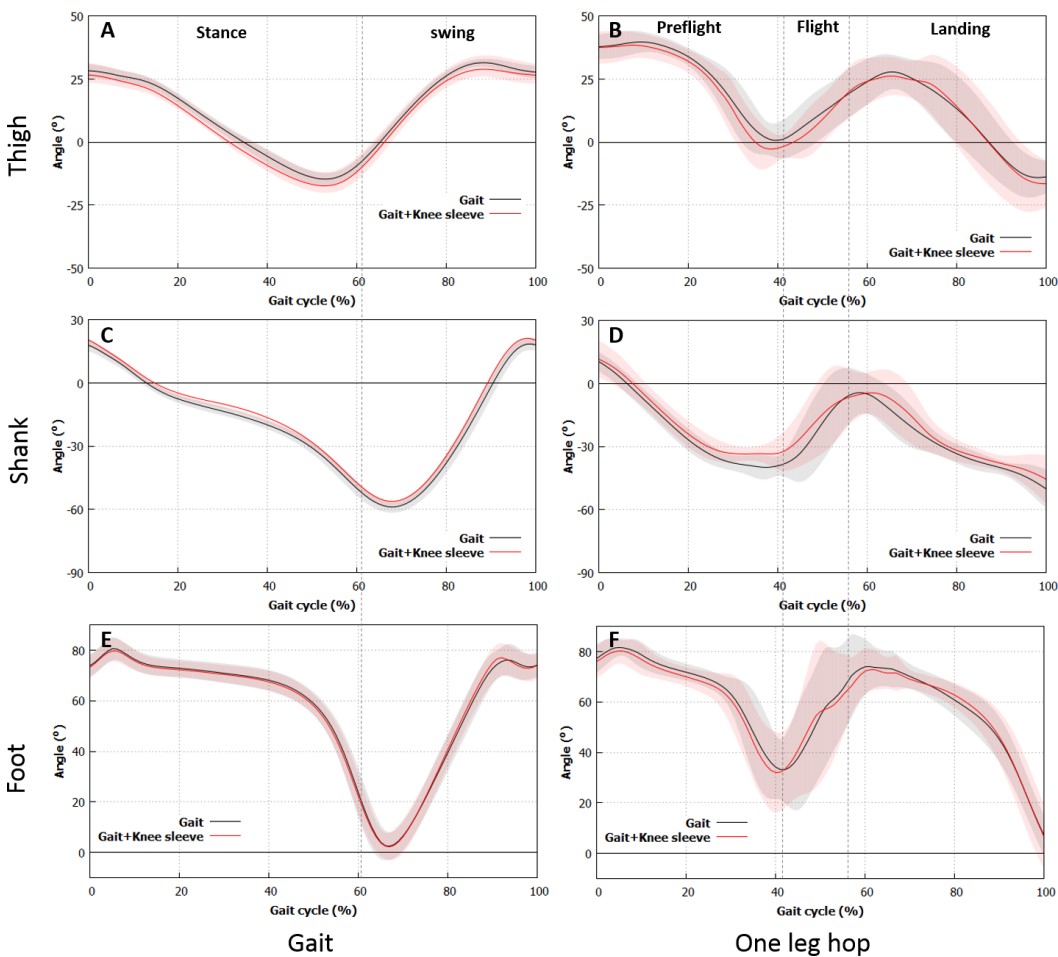

**Figure 2  Variations in the segment angle.** Differences in the segment angles in the thigh and the shank of the subjects with the sleeve and those without are found during gait and one-leg hop. (A), (B) Thigh. (C), (D) Shank. (E), (F) Foot. (A), (C), (E) Gait, (B), (D), (F) One-leg hop.

Figure 3 shows the CRPs, while Tables 3 and 4 present the RMS and SD values of the CRPs, respectively. The RMS value of CRPTS during the stance phase of walking was higher for the subjects while wearing the knee sleeve ($37.3 \pm 4.9$ vs. $39.1 \pm 3.4$, $p = 0.032$), whereas the RMS value of the CRPSF during the stance phase of walking was lower when the subjects wore the knee sleeve ($71.3 \pm 5.9$ vs. $69.0 \pm 6.7$, $p = 0.03$). No differences were found in the other RMS values of the CRPs ($p > 0.05$). The SD value of CRPTS during the swing phase was significantly higher for the the subjects wore the sleeve ($23.5 \pm 5.4$ vs. $28.8 \pm 3.7$, $p = 0.006$). No differences were observed in the other SD values of the CRPs ($p > 0.05$). The RMS values of CRPSF in the preflight and landing phases during the one-leg hop were higher for the subjects wore the sleeve ($88.8 \pm 13.1$ vs. $97.4 \pm 1.8$, $p = 0.010$ and $78.3 \pm 17.5$ vs. $92.2 \pm 2.8$, $p < 0.001$, respectively). The SD values of CRPSF in the preflight phase and of CRPTS and CRPSF in the landing phase were higher for the subjects wore the sleeve ($39.1 \pm 7.2$ vs. $47.3 \pm 7.5$, $p = 0.001$, $49.4 \pm 6.2$ vs. $52.5 \pm 9.7$, $p = 0.043$, and $42.3 \pm 10.0$ vs. $51.4 \pm 13.1$, $p = 0.028$, respectively). No differences were observed in the other SD values of the CRPs ($p > 0.05$).

Ko et al. (2017), *PeerJ*, DOI 10.7717/peerj.3340

**Table 3   Mean ± S.D. values of RMS of CRPs.** The RMS value of CRPTS in the stance during gait is higher for the subjects with the sleeve than those without, whereas that of CRPSF in the stance is lower for the subjects with the sleeve than those without.

| RMS | | CRPTS | | | CRPSF | | | CRPTF | | |
|---|---|---|---|---|---|---|---|---|---|---|
| | | No sleeve | With sleeve | *p*-value | No sleeve | With sleeve | *p*-value | No sleeve | With sleeve | *p*-value |
| Gait | Stance | 37.3 ± 4.9 | 39.1 ± 3.4 | 0.032 | 71.3 ± 5.9 | 69.0 ± 6.7 | 0.030 | 97.8 ± 4.2 | 97.8 ± 5.3 | 0.493 |
| | Swing | 85.4 ± 8.6 | 86.0 ± 7.5 | 0.377 | 30.0 ± 5.0 | 30.9 ± 4.5 | 0.101 | 63.3 ± 10.8 | 62.7 ± 8.6 | 0.395 |
| One-leg hop | Preflight | 78.0 ± 9.8 | 81.0 ± 7.9 | 0.131 | 88.8 ± 13.1 | 97.4 ± 1.8 | 0.010 | 35.4 ± 7.9 | 36.8 ± 6.1 | 0.268 |
| | Flight | 47.3 ± 8.8 | 43.6 ± 7.5 | 0.053 | 27.2 ± 19.6 | 28.9 ± 24.2 | 0.297 | 51.3 ± 8.9 | 54.2 ± 17.4 | 0.268 |
| | Landing | 74.6 ± 18.7 | 87.1 ± 20.9 | 0.065 | 78.3 ± 17.5 | 92.2 ± 2.8 | <0.001 | 56.6 ± 11.4 | 62.8 ± 26.9 | 0.256 |

Ko et al. (2017), *PeerJ*, DOI 10.7717/peerj.3340

**Table 4** **Mean ± S.D. values of SD of CRPs.** The SD value of CRPTS in the swing phase during gait is significantly higher for the subjects with the sleeve than those without. The CRPSF values in the preflight and landing phases during the one-leg hop are higher for the subjects with the sleeve than those without. The SD values of CRPSF in the preflight phase and the SD values of CRPTS and CRPSF in the landing phase are higher for the subjects with the sleeve than those without.

| SD | | CRPTS | | | CRPSF | | | CRPTF | | |
|---|---|---|---|---|---|---|---|---|---|---|
| | | No sleeve | With sleeve | *p*-value | No sleeve | With sleeve | *p*-value | No sleeve | With sleeve | *p*-value |
| Gait | Stance | 29.0 ± 2.8 | 29.6 ± 3.1 | 0.227 | 21.6 ± 2.0 | 20.5 ± 2.8 | 0.100 | 35.4 ± 2.7 | 35.0 ± 2.1 | 0.342 |
| | Swing | 23.5 ± 5.4 | 28.8 ± 3.7 | 0.006 | 20.0 ± 5.1 | 20.9 ± 4.7 | 0.309 | 20.9 ± 4.8 | 22.5 ± 4.6 | 0.143 |
| One-leg hop | Preflight | 36.1 ± 5.2 | 38.3 ± 3.6 | 0.080 | 39.1 ± 7.2 | 47.3 ± 7.5 | 0.001 | 33.1 ± 7.0 | 33.9 ± 5.5 | 0.342 |
| | Flight | 40.3 ± 12.1 | 38.1 ± 13.5 | 0.339 | 17.2 ± 7.6 | 19.0 ± 7.8 | 0.164 | 48.6 ± 13.6 | 47.0 ± 18.0 | 0.370 |
| | Landing | 49.4 ± 6.2 | 52.5 ± 6.7 | 0.043 | 42.3 ± 10.0 | 51.4 ± 13.1 | 0.028 | 50.7 ± 9.1 | 56.1 ± 24.8 | 0.243 |

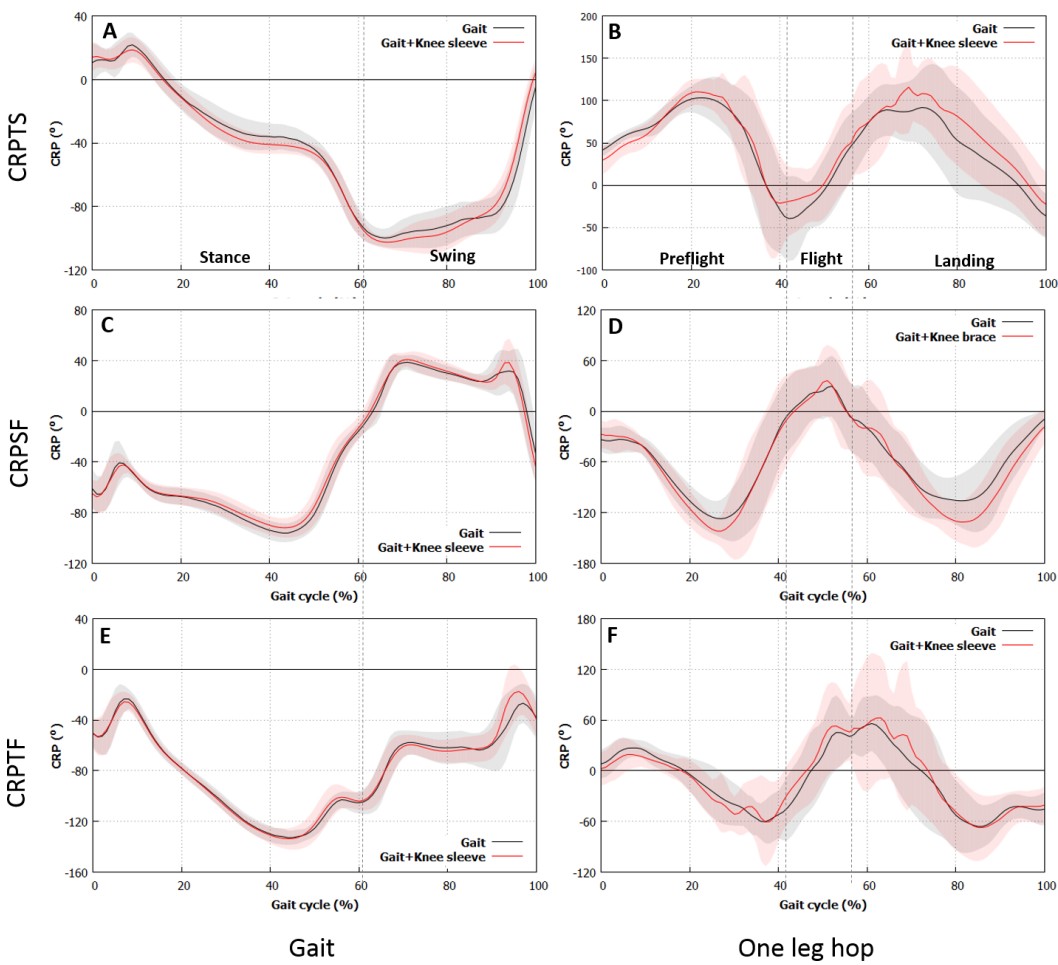

**Figure 3   CRP results.** Differences in CRPTS and CRPSF of the subjects with the sleeve and those without are found during gait and one-leg hop. (A), (B) Thigh. (C), (D) Shank. (E), (F) Foot. (A), (C), (E) Gait, (B), (D), (F) One-leg hop.

# DISCUSSION

The use of a knee sleeve altered the functional capacity and performance associated with joint or segment kinematics. Although the evaluation of the multisegment coordination is more important in gaining a better understanding of the gait and physical activities in humans than that of a single segment, no previous studies have evaluated the effects of a knee sleeve on segment coordination. Therefore, the present study aimed to verify whether the use of knee sleeves affects the coordination of the lower-limb segments during level walking and one-leg hopping.

All the spatiotemporal parameters for the subjects wore the sleeve did not change during the experiments. These results implied that the knee sleeve less altered the subject performance during the gait and the one-leg hop, which was consistent with the inferences made in previous studies (*Bryk et al., 2011*; *Mortaza et al., 2012*).

As regards the segment angle, the initial angle of the thigh when the subjects were wearing the knee sleeve was lower, whereas that of the shank was higher during level
walking. The minimum thigh angle decreased, while the minimum shank angle increased for the subjects with the knee sleeve. These results might imply that wearing the knee brace altered the segment angles during level walking, which was consistent with the observation made by *Collins et al. (2014)*. The knee motion might be restricted by the knee sleeve in the swing phase. However, the differences in the angles of the thigh and the shank between the subjects with and without the knee brace were less than 5°, which was the minimum angle clinically acceptable for identifying the differences between sessions in healthy adults during gait (*Wilken et al., 2012*). Although the angle trajectories for the thigh and the shank completely shifted throughout the level walking activity, the study results implied that wearing the knee sleeve only slightly altered the segment angle. The minimum angles of the thigh and the shank during the one-leg hop were lower (3.5°) and higher (6.3°), respectively, for the subjects with the knee sleeve than those without. No kinematic differences were observed in the foot angles. The minimum angles of the thigh and the shank were normally observed in the late preflight phase. These results indicated that the knee flexion in the late preflight phase was suppressed by the knee sleeve. No alterations were observed in the foot angle during level walking and one-leg hopping. Overall, the effects of the knee sleeve on the segment angles might be dependent on the type of activity: the effects during normal walking were less than those during one-leg hopping.

In terms of the coordinative phase between the segments, the RMS value of CRPTS in the stance phase changed during level walking and particularly increased for the subjects with the knee sleeve. This finding implied a large out-of-phase state between the thigh and the shank (*Lamb & Stöckl, 2014*). In contrast, the RMS value of CRPSF in the stance phase decreased for the subjects with the knee sleeve, thereby implying a large in-phase state between the shank and the foot (*Lamb & Stöckl, 2014*). These results indicated that the knee sleeve might affect the phase between the segments in the stance phase, but not in the swing phase. Although the angle trajectories of each segment between the cases without and with the knee sleeve were similar, the phase differences were observed between each segment. These results might be attributed to the CRP features, in which the direction of the angle movement and the angle velocity were considered (*Lamb & Stöckl, 2014*). The RMS values of CRPSF in the preflight and landing phases during one-leg hopping increased for the subjects with the knee sleeve, indicating an out-of-phase state between the shank and the foot (*Lamb & Stöckl, 2014*). However, no phase changes were observed in the flight phase, which indicated that the knee sleeve might affect the phase between the segments in the preflight and landing phases. Overall, the knee sleeve altered the phase between the segments, particularly in the weight-bearing period or the period of foot contact with the ground in the stance phase during level walking and in the preflight and landing phases during one-leg hopping. These results might be attributed to the proprioceptive enhancement effect of the knee sleeve on the knee (*Barrett, 2003*; *Herrington, Simmonds & Hatcher, 2005*; *Van Tiggelen, Coorevits & Witvrouw, 2008*) and/or the ankle during the weight-bearing period or period of foot contact with the ground to alleviate the load on the knee.

In terms of the coordinative variability between the segments, the variability during tasks could be interpreted in two ways. Greater variability between the segments or joints

was more preferable in the stance phase or the support period, whereas lower variability was more preferable in the swing phase. The lower limb should persistently recognize the terrain to maintain the balance or stability of the body, control the joint positions, and perform tasks (*Yi et al., 2016*) in the stance phase or the support period. Therefore, greater variability was more preferable. Greater variability would also contribute to the load distribution or force on the joint and tissues or suppress the load concentration or force in a small area, thereby leading to the prevention of overuse injuries (*Hamill, Palmer & Emmerik, 2012*). Meanwhile, lower variability might be beneficial in the swing phase to reduce the effort of controlling joints (*Chen, Lu & Chou, 2015*). The SD value of CRPTS in the swing phase changed during level walking (i.e., it particularly increased for the subjects with the knee sleeve, implying a greater variability in the knee for the subjects with the knee brace in the swing phase. Therefore, more coordination patterns should be employed to control the limb during the swing phase of gait (*Chen, Lu & Chou, 2015*). This result might be attributed to a tendency to restrict knee motion in the swing phase, as mentioned above. The influence of variability in the swing phase or the unsupported period on injuries was not studied. Hence, whether the higher variability in the swing phase altered the occurrence of injuries was unclear.

The SD value of CRPSF in the preflight phase and the SD values of CRPTS and CRPSF during landing increased during the single leg hops. The SD value of the joint increased when the knee sleeve was worn, thereby indicating an increase in the joint variability (*Hamill, Palmer & Emmerik, 2012*). Furthermore, the joint type altered by the knee sleeve was likely to be dependent on the activity conditions: the ankle in the preflight phase and the ankle and the knee in the landing phase. The ankle played the most critical role in maintaining balance during the single-leg stance (*Riemann, Myers & Lephart, 2003*). Meanwhile, the proximal joints played an increased role in maintaining balance under more challenging conditions, such as form surface and removal of vision (*Riemann, Myers & Lephart, 2003*). Therefore, wearing the sleeve could improve the subjects' abilities to dynamically balance themselves during a single-leg stance through an increase in the stability of the proximal joints (i.e., knee) that play critical roles in maintaining balance during tasks, such as single leg hopping. In addition, the knee and ankle stability in the landing and preflight phases might be enhanced. Therefore, wearing the knee sleeve could contribute to the prevention of overuse injuries in the knee and/or the ankle via enhancement of the joint stability (*Hamill, Palmer & Emmerik, 2012*).

Injuries frequently occur when a player is in a fatigued state (*Van Tiggelen, Coorevits & Witvrouw, 2008*), and they occur in many sports activities. Therefore, further studies, such as those about athletes, should consider these issues through our evaluation methods to know how a knee sleeve affects the segment kinematics and coordination. In addition to the kinematic analysis, the assessment of the kinetic parameters, such as moment or force, is also needed to evaluate the joint or segment loading or force and understand the effects of knee sleeves on joint mechanics and loading and how a knee sleeve may affect the risk of injury. Several studies found that the knee sleeve might be less effective in young healthy subjects (*Bottoni et al., 2013*) and might produce inhibitory effects in subjects, who are not accustomed to wearing it (*Baltaci et al., 2011*). The subjects in the present study are young

and healthy and not accustomed to wearing the knee sleeve. Nevertheless, the knee sleeve less altered the jumping (i.e., hop distance) and gait performance However, we did not evaluate the potential effect of an acclimation period of wearing the knee sleeve, and how this acclimation may potentially affect their results. Further studies to address these issues are required.

## CONCLUSION

This study aimed to verify whether the use of knee sleeves affects the coordination of lower-limb segments during level walking and one-leg hopping. Herein, the knee sleeve had no significant effects on the jumping and gait performance. In addition, the knee sleeve affected the kinematics of the thigh and the shank, but not of the foot, during level walking and single leg hops. Moreover, the knee sleeve altered the segment coordination in the lower limb during the gait and single leg hop tasks.

### Funding
This research project was supported by the Sports Promotion Fund of Seoul Olympic Sports Promotion Foundation from Ministry of Culture, Sports and Tourism. The funders had no role in study design, data collection and analysis, decision to publish, or preparation of the manuscript.

### Grant Disclosures
The following grant information was disclosed by the authors:
Sports Promotion Fund of Seoul Olympic Sports Promotion Foundation.

### Competing Interests
The authors declare there are no competing interests.

### Author Contributions
- Chang-Yong Ko conceived and designed the experiments, performed the experiments, analyzed the data, contributed reagents/materials/analysis tools, wrote the paper, prepared figures and/or tables, reviewed drafts of the paper.
- Yunhee Chang conceived and designed the experiments, performed the experiments, analyzed the data, contributed reagents/materials/analysis tools, wrote the paper, reviewed drafts of the paper.
- Bora Jeong conceived and designed the experiments, performed the experiments, analyzed the data, contributed reagents/materials/analysis tools, reviewed drafts of the paper.
- Sungjae Kang and Gyoosuk Kim conceived and designed the experiments, performed the experiments, reviewed drafts of the paper.
- Jeicheong Ryu conceived and designed the experiments, performed the experiments, reviewed drafts of the paper, principal investigator for funding.

## Human Ethics

The following information was supplied relating to ethical approvals (i.e., approving body and any reference numbers):

Human Ethics Committee of the Rehabilitation and Engineering Research Institute (RERE-IRB-20160721).

## Data Availability

The raw data has been supplied as a Supplementary File.

## Supplemental Information

Supplemental information for this article can be found online at http://dx.doi.org/10.7717/peerj.3340#supplemental-information.

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
