# Peer review of "Effects of knee sleeves on coordination of lower-limb segments in healthy adults during level walking and one-leg hopping"

_PeerJ, doi:10.7717/peerj.3340_

## Round 0.1 · original submission · Major Revisions

The reviewers seem unanimous in their interest in the findings of the research but also in the areas for improvement which include, 1. strengthening the rationale for the study with a clear research question, 2. providing more methodological details and 3. outlining the limitations of the study.

Please note that Reviewer 1 has also provided edits to your manuscript in an annotated file which is attached. I would encourage you to carefully and completely address the reviewers comments.

Reviewer 1 ·

Basic reporting

The article poses an interesting idea of assessing the effects of a knee sleeve on joint coordination patterns during walking and single leg hops. The authors seemed to have done an extensive literature review for this manuscript, yet it seems like they just simply stated what the previous studies have found as it relates to the topic at hand but did not necessarily use this information to provide a comprehensive formulation of the previous work and the underlying rationale for this manuscript. In addition, the authors should include some sort of hypothesis as there currently is no hypothesis listed. Overall, there are a few grammatical issues that are noted in the attached reviewer edits. One major concern is the figures listed within the text do not match up with the actual figures listed at the end of the paper. The attached edits to the manuscript will provide specific details in regards to the above mentioned comments.

Experimental design

The research question is still somewhat unknown to me and I believe that this lies in the fact that the current introduction is simply a listing of facts (i.e. previous work) but lacks an actual compilation of these facts into a clearly defined research question. Also, please provide a hypothesis for this study.
The experimental methods listed in this manuscript are extremely weak. There is not enough detail for me to go into a motion lab and successfully reproduce this data acquisition. More specific comments can be found in the manuscript edits but some of my major concerns fall within: data collection, data processing and controlling for gait speed.

Validity of the findings

The statistical analyses are sufficient for this type of experimental design. The authors performed a Wilcoxon signed rank test to assess differences in data that were not normally distributed. Please specify the statistical test or method used to assess for uniformity of data.
I would like the authors to consider the fact of whether or not an acclimation period with the knee sleeve would have affected the results of the study. Please discuss this in the discussion, possibly as a limitation of the study.
Also, the figure and table captions do not provide enough information as to what is being depicted in the figures and tables. Please provide more descriptive captions for each of your figures and tables.

Additional comments

I can tell that the authors put in a good effort into performing this study and writing the manuscript. I ask the others to carefully consider all of the feedback provided by the reviewer and to work on improving this manuscript.

Annotated reviews are not available for download in order to protect the identity of reviewers who chose to remain anonymous.

Reviewer 2 ·

Basic reporting

First serious concern is rationale of this study. Especially, writing techniques in the Introduction is critical. The authors referenced a lot of previous literatures, but it does not logically connected towards the aim of this study. For example, the authors cited several references in second and third paragraph of Introduction. However, the sentences contains same references in one sentence (e.g. ‘Chiu et al. (2015) analyzed the changes in interjoint coordination with age during stair walking (Chiu et al. 2015b)’.). Furthermore, current citation style does not mention the DETAILED RESULTS in the previous literatures. Consequently, these paragraph looks redundant, and the massages in each paragraphs are quite weak. Additionally, there was no statement of hypothesis of this study. The authors must develop a lucid and clear hypothesis as to what was expected and why.

Experimental design

Descriptions about experimental procedures and data analysis is redundant and unclear. What was the cut-off frequency of kinematic and kinetic data (and how determined it)? Further, one of the other considerable concerns regards the proper fitting of the braces. Did the authors have a “one-size-fits-all” model for the knee sleeves? This is a severe limitation on the onset. The authors should mention this as part of their discussion, but it should have been an integral part of the study design since an improperly fitted brace would surely bias the results. This is a major flaw that the authors should clarify and keep in mind in future experimental constructs.

Validity of the findings

The results seems to be valid. However, improper fitting of knee sleeves may have postential bias for the results.

Additional comments

In the abstract, although the authors states as ‘wearing a knee sleeve might be an effective technique to prevent injuries,…’ there was no clear descriptions how knee sleeves contribute to the injury prevention (also, it is unclear what type of injuries the authors expected). Indeed, as the authors stated in lines 256-261, the most of injuries in sports activity could be occurred in fatigued situations and unexpected movements. Therefore, current results have strong limitations to suggest actual injury prevention program in athletes. In addition, the authors recruited eleven male subjects with asymptomatic back, hip, knee, and ankle functions. This could also be limitation for proposing injury prevention through knee sleeves.

---

## Round 0.2 · Minor Revisions

The reviewer is satisfied with the improvements made to the article however there are some minor revisions to be made. These are detailed in the Annotated manuscript. Please download from the link at the bottom of this letter.

Reviewer 1 ·

Basic reporting

This is an improvement on the initial submission. The background is improved and the literature cited within the introduction is adequately used.

Experimental design

The research question is much more defined and stated. The methods are more detailed and improved, compared to the initial submission.

Validity of the findings

The results seem valid and justified. The statistics used are sound as well. Conclusions to the original question are sufficient and supported by the results of the study.

Additional comments

The reviewer appreciates the revisions made by the authors. This version of the manuscript is an improvement on the initial submission. Please see the attached document for more specific comments.

Annotated reviews are not available for download in order to protect the identity of reviewers who chose to remain anonymous.

---

## Round 0.3 · accepted · Accept

All the reviewers concerns have been addressed in this version.